

# Offshore Reanalysis Wind Speed Assessment Across the Wind Turbine Rotor Layer off the United States Pacific Coast

Lindsay M. Sheridan[1], Raghu Krishnamurthy[1], Gabriel García Medina[1], Brian J. Gaudet[1], William I. Gustafson[1], Alicia M. Mahon[1], William J. Shaw[1], Rob K. Newsom[1], Mikhail Pekour[1], and Zhaoqing Yang[1]

[1]Pacific Northwest National Laboratory, Richland, WA, USA

*Correspondence to*: Lindsay M. Sheridan (lindsay.sheridan@pnnl.gov)

**Abstract.** The California Pacific coast is characterized by considerable wind resource and areas of dense population, propelling interest in offshore wind energy as the United States moves toward a sustainable and decarbonized energy future. Reanalysis models continue to serve the wind energy community in a multitude of ways and the need for validation in locations where observations have been historically limited, such as offshore environments, is strong. The U.S. Department of Energy (DOE) owns two lidar buoys that collect wind speed observations across the wind turbine rotor layer along with meteorological and oceanographic data near the surface to characterize the wind resource. Lidar buoy data collected from recent deployments off the northern California coast near Humboldt County and the central California coast near Morro Bay allow for validation of commonly used reanalysis products. In this article, wind speeds from the Modern-Era Retrospective Analysis for Research and Applications version 2 (MERRA-2), the Climate Forecast System version 2 (CFSv2), the North American Regional Reanalysis (NARR), the European Centre for Medium-Range Weather Forecasts Reanalysis version 5 (ERA5), and the analysis system of the Rapid Refresh (RAP) are validated at heights within the wind turbine rotor layer ranging from 50 m to 100 m. The validation results offer guidance on the performance and uncertainty associated with utilizing reanalyses for offshore wind resource characterization, providing the offshore wind energy community with information on the conditions that lead to reanalysis error. At both California coast locations, the reanalyses tend to underestimate the observed rotor-level wind resource. Occasions of large reanalysis error occur in conjunction with wind ramps, stable atmospheric conditions, wind speeds associated with peak turbine power production (> 10 m s$^{-1}$), and mischaracterization of the diurnal wind speed cycle in summer months.

## Copyright Statement



## 1 Introduction

As countries around the globe endeavour to decarbonize their economies, offshore wind is gaining momentum to help achieve
targeted emission goals. In 2020, the cumulative global offshore wind capacity stood at 33 GW, with Europe leading the world
in offshore deployment, followed by China (Musial et al., 2021). In North America, offshore wind is still nascent and only
recently became an important resource of focus. This is primarily due to improvements in rates of return, technological
efficiencies, transmission, and confidence stemming from European success (Dong et al., 2021). In 2020, the U.S. offshore
wind project development and operational pipeline stood at a potential generating capacity of 35 GW (Musial et al., 2021).
While much of the offshore wind project development and operational pipeline activity is currently concentrated along the
Atlantic coast, enthusiasm for harvesting the wind resource along the Pacific coast is building (Wang et al., 2019).

Stemming from their essential role in the onshore wind market, reanalysis models are supporting global offshore wind
development and operation in a multitude of ways. These decades-long meteorological and climate data assimilation products
are utilized by the offshore wind community to produce site assessments (Nezhad et al., 2021), simulate long-term power
generation (Hayes et al., 2021), and provide levelized cost of energy estimates (de Assis Tavares et al., 2022). Given their
crucial position in the offshore wind development and operation cycles, validation of reanalyses is imperative to instil and
maintain investor confidence and to set appropriate expectations of power production for load-balancing operatives.

Validations of reanalysis-based wind resource assessments in an offshore wind energy context have been limited in the past
due to a scarcity of observations, particularly across typical turbine rotor layer heights, which range from approximately 25 m
to 200 m for current turbine technology and are projected to range from approximately 25 m to 275 m for future turbine
technology (Musial et al., 2019). Recent advancements in floating lidar technology, however, have propelled opportunities to
enhance understanding of the marine boundary layer while providing the necessary rotor-level height observations for
comparisons with commonly employed reanalyses. Using a lidar system off the coast of India, Jani et al. (2019) noted that the
European Centre for Medium-Range Weather Forecasts (ECMWF)-Interim reanalysis underestimates observed 80 m and 100
m above sea level (ASL) wind speeds by 1.2 m s$^{-1}$. Comparing the ECMWF Reanalysis v5 (ERA5) with observations from a
meteorological tower aboard an oil platform, Fernandes et al. (2021) found zero bias, a root-mean-square error (RMSE) of 2.3
m s$^{-1}$, and a Pearson correlation coefficient of 0.8 at 100 m ASL off the coast of Brazil. In the United States, Pronk et al. (2021)
utilized a floating lidar off the coast of New Jersey and note that ERA5 produces rotor-level biases near -1 m s$^{-1}$, centred root-
mean-square errors (CRMSEs) around 1.5 m s$^{-1}$, and squared correlation coefficients near 0.9 at heights ranging from 58 m to
198 m ASL.

The U.S. Department of Energy (DOE) lidar-mounted buoys deployed for a year off the coasts of New Jersey and Virginia
showed consistent reanalysis and model underestimation of wind speed at 90 m ASL (Shaw et al., 2020; Sheridan et al., 2020).
For these Atlantic locations, ERA5 and Rapid Refresh (RAP) showed high correlation with the 90 m wind speed observations
(Pearson correlation coefficient ~0.9) and a RMSE of ~1.9 m s$^{-1}$. Significant reanalysis underestimation and reduced
correlation was noted during the summer at the New Jersey and Virginia buoy locations (Sheridan et al., 2021). Atlantic



meteorological and synoptic conditions such as sea breezes, tropical storms, and coastal upwelling and downwelling conditions were identified as some of the typical causes for reanalysis-based wind speed errors (Sheridan et al., 2021).

In this article, year-long measurements from two buoy deployments off the coast of California are analysed. The wind resource along the Pacific coast of California is influenced by diverse physical characteristics, such as low-level jets (Parish, 60 2000), coastally trapped wind reversals (Bond et al., 1996), atmospheric hydraulic jumps (Juliano et al., 2017), Santa Ana winds (Rolinski et al., 2019), and California expansion fans (Parish et al., 2016). The marine boundary layer along the western coast is heavily influenced by the coastline. The points and capes near the two lidar buoys (Cape Mendocino near Humboldt and Point Conception near Morro Bay) significantly impact the marine boundary layer. These features influence the wind and temperature fields creating stronger gradients along the coast. These conditions can also result in horizonal ducting, which 65 influences microwave transmission within the region (Haus et al., 2021). Fog and low-level stratus are also frequently observed along the California coast (Koracin et al., 2014). Fog formation within the region is primarily due to advection of moist air over cold upwelled waters. Fog is observed mostly during summer months. Clouds also influence satellite retrievals of various sea state parameters (such as sea surface temperature), which are typically used in reanalysis models. Lack of long-term offshore boundary layer observations for data assimilation into reanalysis models results in larger errors in offshore conditions. 70 In general, complex conditions are observed along the coast of California and modelling of the wind resource remains a continuous challenge to the research community.

Analysis of the wind observations collected during the California deployments of the DOE lidar buoys is provided in Sect. 2, along with descriptions of the models and satellite observations used in this study. Sect. 2 concludes with a discussion of the error metrics selected to validate each reanalysis and analysis model. Sect. 3 examines the overall performances of the 75 models during the California deployment period and continues with a breakdown of performance according to a variety of temporal and physical characteristics, such as trends according to seasonal and diurnal cycles, atmospheric stability, wind reversals, ramp events, and case studies on major atmospheric phenomenon observed during the buoy deployments. Finally, Sect. 4 summarizes the results to provide an understanding of the capabilities and limitations associated with reanalysis-based wind resource assessment in California coastal areas of offshore wind development interest.

**2 Data Discussion and Methodology**

The U.S. DOE buoys underwent extensive upgrades in 2019 and were retrofitted with enhanced Doppler lidar systems and other meteorological sensors (Severy et al., 2021). Prior to their deployment off the California coast, the DOE lidar buoys underwent extensive validation administered by Ocean Tech Services and Det Norske Veritas (DNV) at the Martha's Vineyard Coastal Observatory operated by Woods Hole Oceanographic Institute in the spring of 2020. The validation was performed 85 against an International Electrotechnical Commission (IEC)-certified reference lidar approximately 250 m away on an offshore platform (Air-Sea Interaction Tower). The comparison yielded wind speed coefficients of determination ($R^2$) greater than 0.98 and wind direction $R^2$ values greater than 0.97 at heights up to 200 m ASL (Gorton and Shaw, 2020).



In the fall of 2020, the buoys were deployed off the northern and central coasts of California (Fig. 1). The central buoy was

deployed in 1100 m of water depth approximately 40 km from the shore near Morro Bay (35.7107°N, 121.8581°W). The Morro Bay buoy was deployed from 29 September 2020 to 19 October 2021. The northern buoy was deployed in 625 m of water approximately 40 km off the coast of Humboldt County (40.97°N, 124.5907°W). The Humboldt buoy deployment period began 1 October 2020 and is estimated to conclude in May 2022. A large wave event in December 2020 disabled the buoy power supply, which resulted in a significant gap in the data availability during the first year of deployment (Fig. 2d). In this

article, the period of study for both buoys is 1 October 2020 to 30 September 2021.

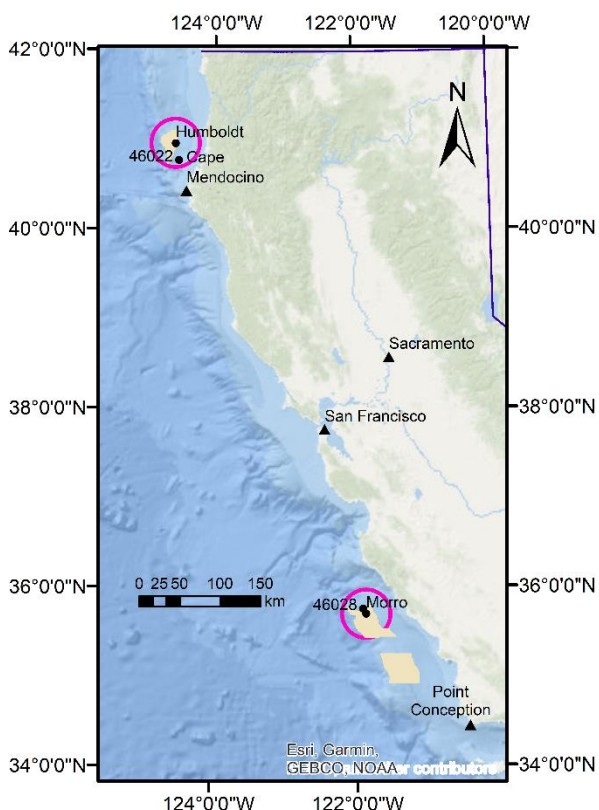

**Figure 1.** Locations of the DOE buoys and nearby National Data Buoy Center buoys 46022 and 46028. The shaded regions around the buoys show Bureau of Ocean Energy Management wind energy areas as of 12 January 2022.

**2.1 Buoy Instrumentation and Observations**

The U.S. DOE owns two AXYS WindSentinel™ buoys mounted with Leosphere WindCube V2 lidar systems and surface meteorological and oceanographic (metocean) instruments. During the California deployments, the instruments were identical on both buoys as listed in Table 1. The buoys collect metocean observations including wind speed and direction from the



surface up to 250 m ASL and current profiles down to 200 m below the sea surface. A complete description of the instrumentation aboard the buoys is provided in Severy et al. (2021).


**Table 1. Description of instrument manufacturer and models.**

| Sensor Type | Make/Model | Measurements |
|---|---|---|
| Wind profiling lidar | Leosphere/WindCube 866 | Vertical profile of motion-compensated wind speed and direction, wind dispersion, and spectral width |
| Cup anemometer | Vector Instruments/A100R | Horizontal wind speed, near surface |
| Wind vane | Vector Instruments/WP200 | Horizontal wind direction, near surface |
| Ultrasonic anemometer | Gill/WindSonic | 2D wind velocity and direction, near surface |
| Pyranometer | Licor/LI-200 | Global solar radiation |
| Temperature | Rotronic/MP101A | Air temperature |
| Relative humidity | Rotronic/MP101A | Relative humidity |
| Acoustic Doppler current profiler | Nortek/Signature 250 | Ocean current speed and direction from sea surface to 200 m water depth |
| Conductivity temperature depth | Seabird/SBE 37SMP-1j-2-3c | Conductivity and sea surface temperature |
| Directional wave sensor | AXYS/TRIAXYS NW II | Directional wave spectra, wave height, and wave period |
| Water temperature | AXYS/YSI | Sea surface temperature |
| Buoy built-in inertial motion unit (for wind vane correction) | MicroStrain/3DM GX3 25 | Yaw, pitch, roll, and global position |
| DOE inertial motion unit (for Doppler lidar motion compensation) | MicroStrain/3DM GX5 45 | Yaw, pitch, roll, linear velocity, global position, magnetometer, and gyroscope |

The winds at the buoy deployment locations predominantly follow the California coastline, northerly at Humboldt (Fig. 2b) and northwesterly at Morro Bay (Fig. 2e). The average wind speed at 100 m ASL is 9.0 m s$^{-1}$ over the Humboldt deployment and 8.6 m s$^{-1}$ over the Morro Bay deployment. Applying the reference 6 MW power curve of Musial et al. (2019), the 100 m wind speeds translate to deployment-wide gross capacity factors (the ratio of simulated deployment-wide energy in kWh with





the product of turbine rated capacity and the number of hours in the deployment) of 59.6% at Humboldt and 55.8% at Morro
Bay (without taking into effect any operational losses, such as wakes, turbine availability, etc.).

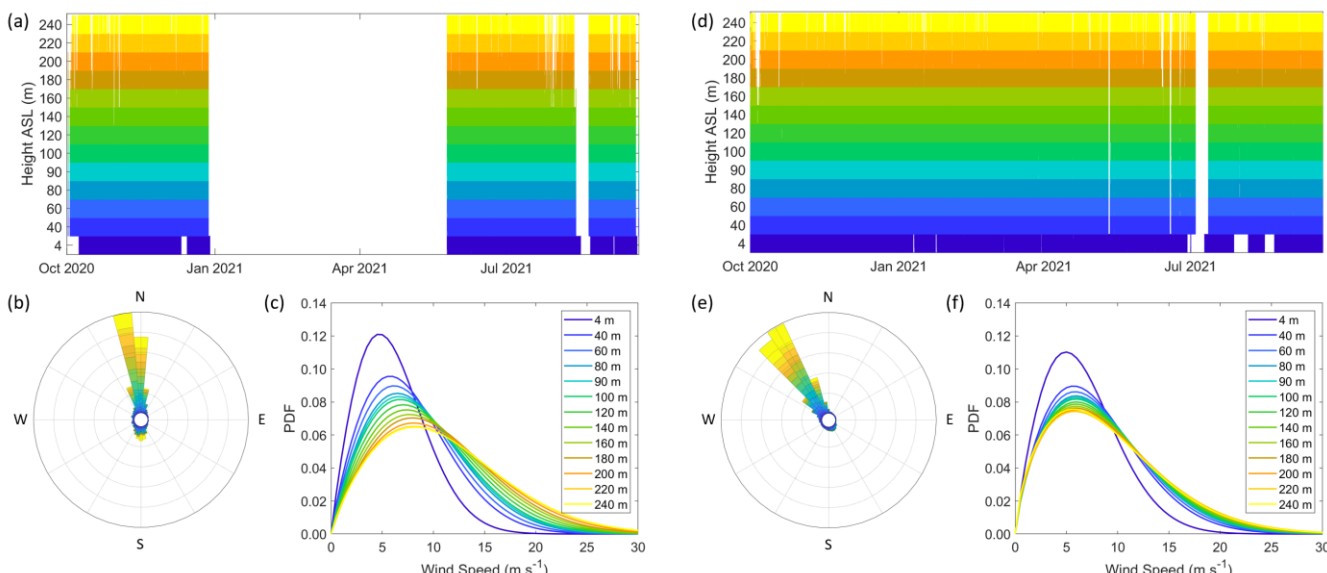

**Figure 2.** Observational coverage of the (a) Humboldt and (d) Morro Bay deployments, observed 100 m wind roses from the (b) Humboldt
and (e) Morro Bay deployments, Weibull fits to wind speed observations from the (c) Humboldt and (f) Morro Bay deployments.


## 2.2 Reanalysis and Analysis Models

Five reanalysis and weather forecast analysis models are investigated for their performance in representing the offshore wind
resource at the lidar buoy locations off the California coast. MERRA-2 from the National Aeronautics and Space
Administration (NASA) Global Modeling and Assimilation Office (GMAO) is a global reanalysis that covers the modern
satellite era (Gelaro et al., 2017). The Climate Forecast System version 2 (CFSv2) from the National Oceanic and Atmospheric
Administration (NOAA) National Centers for Environmental Prediction (NCEP) is a global operational forecast model that
runs four times daily and is extended temporally by its reanalysis component, the Climate Forecast System Reanalysis (CFSR)
(Saha et al., 2011, 2014). NARR from NOAA NCEP is a reanalysis with spatial coverage over North America and temporal
coverage over the modern satellite era (Mesinger et al., 2006). ERA5 from ECMWF is a global reanalysis with the longest
temporal coverage of the five models assessed in this work, extending from 1950 (Hersbach et al., 2020). RAP is an operational
forecast model covering North America with a temporal extent covering the last 10 years (Benjamin et al., 2016). The analysis
system of RAP is considered in this study. Since this study uses a combination of analysis and reanalysis data, we will refer to
these collectively as model data henceforth for simplification. Spatial and temporal characteristics of the five models are
provided in Table 2.





**Table 2.** Characteristics of the models that produce rotor-level wind speed estimates.

| Model | MERRA-2 | CFSv2 | NARR | ERA5 | Rapid Refresh |
|---|---|---|---|---|---|
| Developer | NASA GMAO | NOAA NCEP | NOAA NCEP | ECMWF | NOAA NCEP |
| Temporal Coverage | 1980 – present | 2011 – present[a] | 1979 – present | 1950 - present | 2012 – present |
| Temporal Output Frequency | 1-hr | 1-hr | 3-hr | 1-hr | 1-hr |
| Spatial Coverage | Global | Global | North America | Global | North America |
| Horizontal Grid Spacing | 0.5° x 0.625° | 0.5°[b] | 32 km | 0.5°[c] | 13 km |
| Wind Speed Output Near Surface, Rotor-Level Heights | 2 m, 10 m, 50 m | 10 m, lowest 0-30 mb layer ASL | 10 m, Lowest 0-30 mb layer ASL | 10 m, 100 m | 10 m, 80 m, lowest 0-30 mb layer ASL |
| Nearest Grid Point to Humboldt Buoy | 19 km to the east-northeast | 9 km to the east-northeast | 19 km to the north | 9 km to the east-northeast | 5 km to the west-southwest |
| Nearest Grid Point to Morro Bay Buoy | 24 km to the south | 28 km to the southwest | 9 km to the west-northwest | 11 km to the east-northeast | 2 km to the northeast |

[a] 1979 – 2010 is available as CFSR.
[b] The data have been converted from the native reduced Gaussian grid to a regular latitude-longitude grid at 0.5° (Saha et al., 2011).
[c] The data have been converted from the native reduced Gaussian grid to a regular latitude-longitude grid at 0.25° (Hersbach et al., 2020).

Several of the models provide wind data at turbine hub heights in support of the wind energy industry (Ramon et al., 2019). MERRA-2 outputs wind data at 50 m, which are interpolated from the lowest native level using the Helfand and Schubert scheme (Helfand and Schubert, 1995). The Helfand and Schubert scheme uses modified Monin-Obukhov similarity theory (MOST) with roughness length parameterized as in Large and Pond (1981). RAP outputs wind data at 80 m, which are determined by interpolation between the surrounding prognostic model levels (Benjamin et al., 2020). ERA5 outputs wind data at 100 m, which are derived using MOST with open-terrain roughness (Ramon et al., 2019). Additionally, three models (RAP, CFSv2, and NARR) output wind data at the lowest 0-30 mb layer ASL. This variable represents the average wind speed over the layer from the surface to the height at which the air pressure is equal to the surface pressure minus 30 mb. This output layer is frequently used by the industry to be representative of hub height winds. Therefore, a thorough evaluation of this output with lidar measurements at various altitudes would help the wind industry tailor their wind resource expectations from the model.



**Figure 3.** Average wind speed over the period 1 October 2020 through 30 September 2021 from (a) MERRA-2 at 50 m ASL, (b) CFSv2 over the lowest 0-30 mb layer ASL, (c) NARR over the lowest 0-30 mb layer ASL, (d) ERA5 at 100 m, (e) RAP at 80 m, and (f) RAP over the lowest 0-30 mb layer ASL. Buoy deployment locations are indicated with stars. Model grid points are indicated with dots.





## 2.3 Near Surface Wind Data

In order to provide a more comprehensive analysis of the wind off the California coast, two additional data sources providing near surface wind speed data are considered. The NOAA National Data Buoy Center (NDBC) maintains historical and current observational data from a network of buoys across U.S. waterways (NDBC, 2021). Two NDBC buoys are located near the DOE lidar buoy deployment locations at Humboldt and Morro Bay, each of which provide wind speed information at 4 m ASL, which is consistent with the anemometer height aboard the DOE buoys. NDBC buoy 46022 is located at 40.748°N, 124.527°W, approximately 25 km south-southeast of the DOE Humboldt buoy, and provides wind speed observations with a data recovery of 91% over the period 1 October 2020 through 30 September 2021. NDBC buoy 46028 is located at 35.77°N, 121.903°W, approximately 8 km northwest of the DOE Morro Bay buoy, and provides wind speed observations with a data recovery rate of 96% over the period 1 October 2020 through 30 September 2021.

Ribal and Young (2019) provide a 33-year collection of satellite-derived 10 m wind speeds from 14 altimeters with global coverage. Satellite near surface wind data from CRYOSAT-2, JASON-3, SARAL, SENTINEL-3A, and SENTINEL-3B with a quality control flag of 1, indicating good data, are utilized in this study, and the uncalibrated version is chosen in order to align with the data that is assimilated into the reanalyses. The satellite data offer sporadic near surface wind speed measurements because satellites follow multi-day repeat tracks. The temporal resolution along the tracks is high, approximately 1 Hz (Ribal and Young, 2019). 627 and 203 satellite data points are collected within a 30-km radius of the Humboldt and Morro Bay buoys, respectively, at times across the diurnal cycle. Seasonally, satellite data representation is balanced across all months between October 2020 and September 2021, except July, August, and September, when little to no satellite data is available due to possible presence of low-level clouds or fog.

## 2.4 Atmospheric Stability Data

Atmospheric stability is a major influence on the shape of the vertical wind speed profile, and therefore on the amount of power that can be derived from a wind turbine (Wharton and Lundquist, 2012). To assess the impact of atmospheric stability on model wind speed performance, the Obukhov length $L$ is estimated using the Tropical Ocean-Global Atmosphere Coupled-Ocean Atmosphere Response Experiment (TOGA/COARE) version 3.6 algorithm using the near surface DOE buoy metocean observations at Humboldt and Morro Bay (Fairall et al., 1996; Edson et al., 2013). Typically, $L$ is defined as:

$$L = -\frac{\overline{T_v} \cdot u_*^3}{k \cdot g \cdot \overline{w'T_v'}} \tag{1}$$

where $T_v$ is the virtual temperature, $u_*$ is the friction velocity, $k$ is the von Kármán constant, $g$ is gravitational acceleration, and $\overline{w'T_v'}$ is the kinematic virtual temperature flux. In COARE version 3.6, the Obukhov length can be expressed as an effective function of the bulk Richardson number (Grachev and Fairall, 1997). The bulk Richardson number is defined purely using standard meteorological and oceanic mean observations and is given by





$$Ri_b = -\frac{g}{T}\frac{z\Delta\theta_v}{U^2} \tag{2}$$

where, $\Delta\theta_v$ is the virtual potential temperature difference between the water surface and atmosphere, $T$ is the air temperature,
$U$ is the magnitude of the mean wind vector and $z$ is the height of measurement.

## 2.5 Methodology

Given the location of both DOE buoys within or near the dense wind speed contours along the coastline (Fig. 3), selecting the nearest neighbour model grid point does not provide a representative baseline for comparison between simulated and observed
wind speeds. Therefore, distance-weighted interpolation using the surrounding model grid points is applied to approximate the simulated wind speed at the buoy locations. Vertically, models with single level output heights that align with the lidar output heights, i.e., RAP at 80 m ASL and ERA5 at 100 m ASL, are directly compared with the observations at that height. For MERRA-2 at 50 m, a height that does not align with the lidar output heights, the observations at 40 m and 60 m are linearly interpolated to 50 m to provide comparison at that height. For the three models that output wind speed data over the lowest 0-
30 mb layer ASL (CFSv2, NARR, and RAP), comparisons are performed using the lidar buoy observations at all output heights between 80 m and 120 m to determine which hub height(s) the layer best represents.

Temporally, comparisons between observed and model wind speeds are performed according to the resolution of the model, namely on a 3-hourly basis for NARR and a 1-hourly basis for MERRA-2, CFSv2, ERA5, and RAP. The processed lidar buoy observations are averaged over a 10-minute period, with the timestamp reflecting the start of the averaging period. The lidar
buoy observations at the top of each hour are selected for comparison with the models. The NDBC buoy observations are averaged over a 10-minute period, with the timestamp reflecting the end of the averaging period. The NDBC buoy observations at 10 minutes after the hour are therefore selected to align with the averaging period of the lidar buoy observations. Given the temporal coarseness of the satellite winds, wind data from this collection are down-selected within a 30-km radius of the buoys and considered for their average and distribution characteristics.
In order to assess the accuracy of model representation of observed wind speeds, the wind speed validation employs three error metrics: wind speed bias, centred root-mean-square error (CRMSE), and correlation coefficient. The wind speed bias, i.e., the average difference over a timeseries of length $N$ between the modelled ($v_{mod}$) and observed ($v_{obs}$) wind speeds, conveys whether a model tends to overestimate (positive bias), underestimate (negative bias), or accurately represent (zero bias) the observed wind resource:

$$Bias = \frac{1}{N}\sum_{i=1}^{N}\left(v_{mod,i} - v_{obs,i}\right) \tag{3}$$

The CRMSE describes the variation in error between modelled and observed wind speeds, with larger values corresponding to larger error:





$$CRMSE = \sqrt{\frac{1}{N}\sum_{i=1}^{N}\left(\left(v_{mod,i}-\overline{v_{mod}}\right)-\left(v_{obs,i}-\overline{v_{obs}}\right)\right)^2} \tag{4}$$

Finally, the Pearson correlation coefficient describes the degree to which the modelled and observed wind speeds are linearly
related, with values close to one indicating a high degree of correlation:

$$Correlation = \frac{\sum_{i=1}^{N}(v_{mod,i}-\overline{v_{mod}})\,(v_{obs,i}-\overline{v_{obs}})}{\sqrt{\sum_{i=1}^{N}(v_{mod,i}-\overline{v_{mod}})^2}\,\sqrt{\sum_{i=1}^{N}(v_{obs,i}-\overline{v_{obs}})^2}} \tag{5}$$

## 3 Results

### 3.1 Model Wind Speed Performance over the California Deployments

At the Humboldt and Morro Bay sites, the models tend to underestimate the observed rotor-level wind speeds (Fig. 4).
Beginning with the direct model level to lidar level comparisons at Morro Bay, MERRA-2, the coarsest of the models, strongly
underestimates the average observed wind speed at 50 m ASL by 1.6 m s$^{-1}$. Meanwhile at Humboldt, MERRA-2 exhibits
almost no bias at 50 m. ERA5 overestimates the observed 100 m wind speed at Humboldt and underestimates the observed
100 m wind speed at Morro Bay by approximately the same magnitude (0.4 m s$^{-1}$). At 80 m ASL, RAP underestimates with
similar biases at each deployment site, -0.9 m s$^{-1}$ and -0.8 m s$^{-1}$ at Humboldt and Morro Bay, respectively.

Using the lowest 0-30 mb layer ASL to approximate the hub height wind resource produces negative biases at both lidar
buoy sites regardless of the observed hub height selected for comparison (Fig. 4). At each site and for each model (CFSv2,
NARR, and RAP), the average simulated wind speed produced using the lowest 0-30 mb layer is closest to the average
observed wind speed at 80 m. Comparable in horizontal resolution to MERRA-2, CFSv2 similarly produces an 80 m wind
speed bias near zero at Humboldt and a bias of -0.8 m s$^{-1}$ at Morro Bay. NARR underestimates the observed 80 m wind speed
by 0.3 m s$^{-1}$ at Morro Bay and drastically underestimates the observed 80 m wind speed by 1.9 m s$^{-1}$ at Humboldt. Using the
lowest 0-30 mb layer ASL, RAP produces 80 m wind speed biases of -0.8 m s$^{-1}$ and -0.6 m s$^{-1}$ at Morro Bay and Humboldt,
respectively, the latter of which is smaller than the bias produced using the direct RAP 80 m output level at Humboldt.



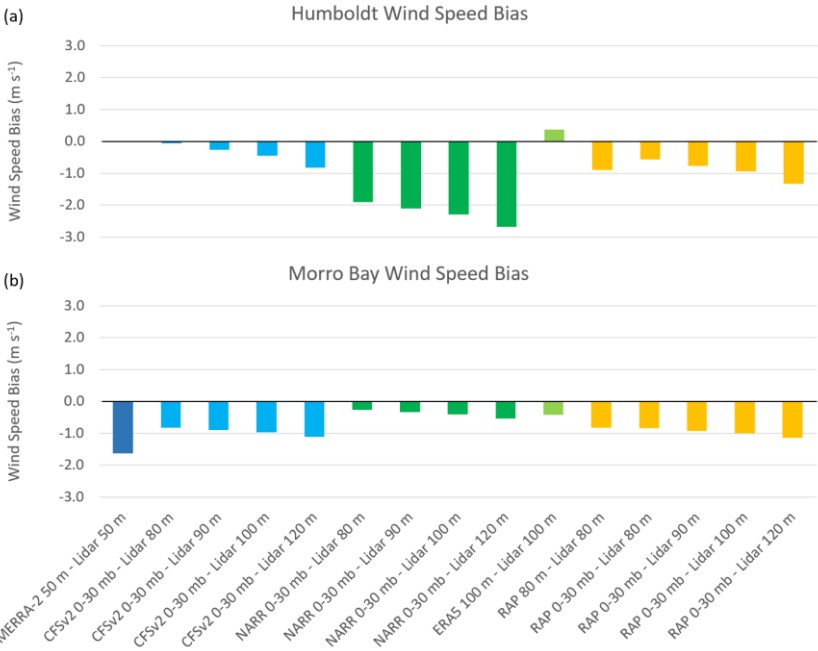

**Figure 4.** Wind speed biases during the (a) Humboldt and (b) Morro Bay deployments.

An examination of modelled and observed wind speed behaviour near the surface, the level at which buoy observations and satellite winds are assimilated into reanalyses, is performed. With the exception of NARR, the models and the satellite winds at 10 m capture the enhanced wind speeds within the expansion fan caused by the bend in the California coastline at 40°N (Fig. 5). MERRA-2, CFSv2, and the satellite winds show two locations of wind speed maxima, to the north and south of the Humboldt buoy, while ERA5 and RAP concentrate the fastest wind speeds to the south of the Humboldt buoy.

The supercritical flow that is developed south of Humboldt at Cape Mendocino results in increasing wind speeds south of the Cape creating an expansion fan (Dorman et al., 1995, Haack et al., 2001). The satellite winds and models show the DOE Humboldt buoy and neighbouring NDBC buoy 46022 located within a swath of rapid wind speed deceleration between the expansion fan zone and the California coast (Fig. 5). Weibull fits to the near surface wind speeds at Humboldt reveal consistent distributions for the buoy observations, satellite winds, and the MERRA-2, CFSv2, ERA5, and RAP wind speeds, with the

buoy observations yielding slightly slower distributions reflective of the lower measurement height of 4 m (Fig. 6a). NARR at 10 m shows substantially slower wind speeds than the observations, satellite winds, and other models at Humboldt, consistent with the trends in rotor layer wind speed bias (Fig. 4a).

      The satellite winds, CFSv2, and ERA5 show a swath of 10 m wind speed between 7.5 m s⁻¹ and 8 m s⁻¹ extending toward the DOE Morro Bay buoy and neighbouring NDBC buoy 46028, at approximately 35.5°N and 122°W (Fig. 5). MERRA-2,

NARR, and RAP place 10 m wind speeds in this range further from shore, and therefore further from the observational buoys. The MERRA-2 wind speed deceleration zone near Morro Bay is particularly large in geographic extent. Weibull fits to the near surface wind speeds at Morro Bay show similar behaviour between CFSv2 and ERA5, with 10 m distributions peaking



around 6 m s$^{-1}$ (Fig. 6b). The NARR and RAP 10 m distributions are similar to the DOE and NDBC buoy distributions at 4 m. MERRA-2 at 10 m shows drastically slower wind speeds than the observations, satellite winds, and other models at Morro

Bay, a finding that aligns with the trends in rotor-level wind speed bias noted in Fig. 4b.



**Figure 5.** Average wind speed over the period 1 October 2020 through 30 September 2021 from (a) MERRA-2 at 10 m ASL, (b) CFSv2 at 10 m ASL, (c) NARR at 10 m ASL, (d) ERA5 at 10 m ASL, (e) RAP at 10 m ASL, and (f) the satellite winds at 10 m ASL. Model grid points are indicated with dots. DOE buoy locations are indicated with stars and are coloured to reflect the average 4 m ASL wind speed over the period 1 October 2020 through 30 September 2021 (6.6 m s$^{-1}$ at the northern Humboldt location and 6.1 m s$^{-1}$ at the central Morro Bay location). NDBC buoy locations are indicated with circles and are coloured to reflect the average 4 m ASL wind speed over the period 1 October 2020 through 30 September 2021 (6.7 m s$^{-1}$ at the northern Humboldt location and 5.8 m s$^{-1}$ at the central Morro Bay location).



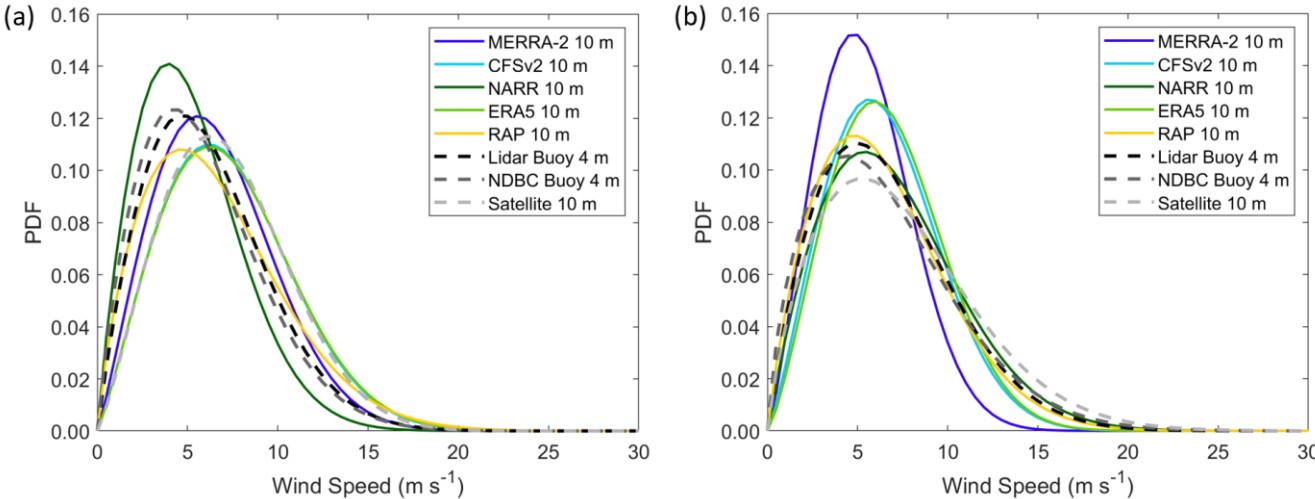


**Figure 6.** Weibull fits to near surface observed, modelled, and satellite winds during the (a) Humboldt and (b) Morro Bay deployments.

For the performance metrics of CRMSE and correlation, RAP is the best performing model at both sites, regardless of whether the direct 80 m output level or the lowest 0-30 mb layer ASL is employed (Fig. 7). Beginning with the direct model

level to lidar level comparisons, RAP, ERA5, and MERRA-2 produce CRMSEs of 2.3 m s⁻¹, 2.4 m s⁻¹, and 2.7 m s⁻¹, respectively, and correlations of 0.88, 0.88, and 0.79, respectively, at Humboldt. RAP, ERA5, and MERRA-2 produce CRMSEs of 1.7 m s⁻¹, 2.3 m s⁻¹, and 2.6 m s⁻¹, respectively, and correlations of 0.94, 0.89, and 0.86, respectively, at Morro Bay.

Simulating wind speeds at 80 m using the RAP, CFSv2, and NARR lowest 0-30 mb layer ASL produces CRMSEs of 2.3

m s⁻¹, 2.8 m s⁻¹, and 2.6 m s⁻¹, respectively, and correlations of 0.88, 0.82, and 0.83, respectively, at Humboldt (Fig. 7). At Morro Bay, the RAP, CFSv2, and NARR lowest 0-30 mb layer ASL produce CRMSEs of 1.8 m s⁻¹, 2.2 m s⁻¹, and 2.3 m s⁻¹, respectively, and correlations of 0.93, 0.90, and 0.89, respectively, compared to observed 80 m wind speeds. The model-based rotor-level correlations during the Humboldt and Morro Bay deployments are similar to or exceed the near surface correlations determined by Wang et al. (2019) comparing NDBC buoy observations along the California coast with NARR and MERRA

(version 1).

While comparing the lowest 0-30 mb layer ASL with different observational heights (80 m – 120 m) produces noticeable variability in the resultant biases (Fig. 4), with the lowest bias occurring when comparing with the observations at 80 m, correlation and CRMSE do not show a trend with varying observational height. Standard deviations in the CRMSEs produced when comparing the lowest 0-30 mb layer ASL with observations at 80 m, 90 m, 100 m, and 120 m range from 0.04 m s⁻¹

(RAP at Humboldt, best case) to 0.1 m s⁻¹ (NARR at Humboldt, worst case). Standard deviations in the correlations produced





when comparing the lowest 0-30 mb layer ASL with observations at 80 m, 90 m, 100 m, and 120 m range from 0.06% (RAP at Morro Bay, best case) to 0.2% (NARR at Humboldt, worst case).

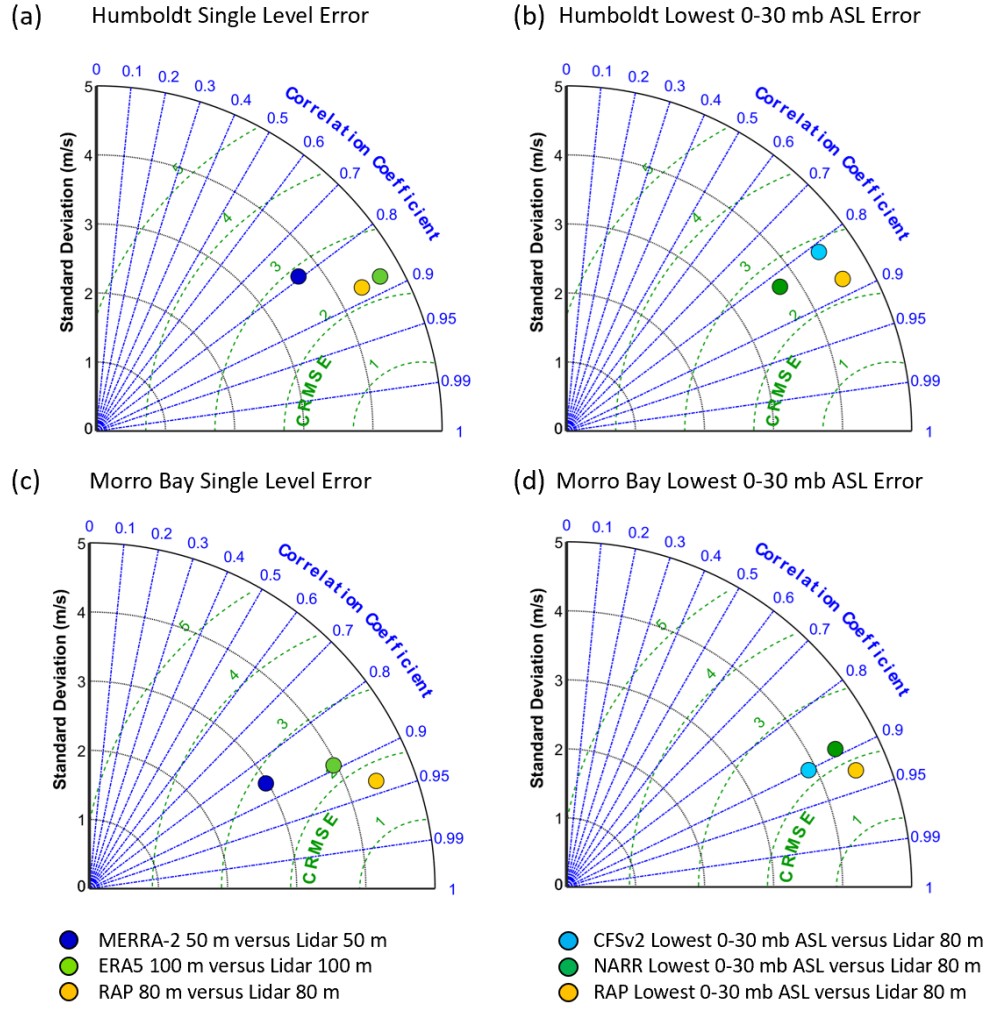

**Figure 7.** Taylor diagrams of single level model error at (a) Humboldt and (c) Morro Bay, and error using the lowest 0-30 mb layer ASL to simulate hub height at (b) Humboldt and (d) Morro Bay.

## 3.2 Model Performance According to Atmospheric Stability

The COARE model in conjunction with the near surface measurements from the DOE lidar buoys enables an examination of model performance according to atmospheric stability. The dimensionless ratio $z/L$ ($z = 4$ m) is concentrated around zero for both the Humboldt and Morro Bay deployments (Fig. 8a, c), indicating a predominance of near neutral atmospheric conditions.



Outside of neutral, Morro Bay conditions tend toward unstable ($z/L < 0$) while the Humboldt conditions are more evenly distributed among unstable and stable ($z/L > 0$).

For both the Humboldt and Morro Bay deployments, the models show the strongest underestimation of the observed rotor-level wind speeds when atmospheric conditions near the surface are near neutral ($z/L \approx 0$). At Humboldt, MERRA-2, CFSv2,
NARR, ERA5, and RAP produce rotor-level wind speed biases of -1.1 m s$^{-1}$, -0.8 m s$^{-1}$, -2.9 m s$^{-1}$, -0.3 m s$^{-1}$, and -1.6 m s$^{-1}$, respectively, when $-0.05 \le z/L < 0.05$ (Fig. 9b). At Morro Bay, MERRA-2, CFSv2, NARR, ERA5, and RAP produce rotor level wind speed biases of -3.0 m s$^{-1}$, -1.6 m s$^{-1}$, -0.8 m s$^{-1}$, -1.3 m s$^{-1}$, and -1.3 m s$^{-1}$, respectively (Fig. 8d).

At Humboldt, the biases in the unstable regime are confined between ±1 m s$^{-1}$, while in the stable regime the biases indicate strong model overestimation of the observed wind speeds, with biases in excess of 2 m s$^{-1}$ for MERRA-2, CFSv2, and ERA5
(Fig. 8b). At Morro Bay, the trend is reversed, as the models most strongly overestimate the observed rotor-level wind speeds in the unstable regime, with all models except RAP exhibiting biases near 1 m s$^{-1}$ (Fig. 8d). As with Humboldt in unstable conditions, the biases at Morro Bay during stable conditions tend to be confined between ±1 m s$^{-1}$ (Fig. 8d). The impact of atmospheric stability is also commonly observed in MOST, which typically shows larger errors during stable atmospheric conditions. Since models use MOST to extrapolate within the lowest grid levels, the propagation of error in the models is
dependent on the accuracy of the chosen MOST, such as Businger (1971) and Dyer (1974), Beljaars and Holtslag (1991), and Vickers and Mahrt (1999), in the region of interest. Since both Humboldt and Morro Bay are considerably offshore, MOST is expected to be valid, but the model deviations can be higher depending on the chosen similarity model and observed sea state conditions. At Humboldt, nonstationary conditions and high waves (maximum wave height of 40 feet on Dec 7, 2020) are observed which can result in conditions that are not suitable for MOST within the surface layer. The marine atmospheric
boundary layer height also plays a significant role, as MOST is generally valid only within the surface layer (10 % of the atmospheric boundary layer). Due to strong gradients in boundary layer depth at these two sites, presence of nearby points and capes, the applicability of MOST varies drastically in height along the California coast.

Atmospheric stability is highly correlated with wind shear and turbulence intensity (Wharton and Lundquist, 2012). High wind shear is typically associated with stable atmospheric stratification while low shear is associated with strongly convective
atmospheric conditions. Similarly, turbulence intensity (defined as the ratio of standard deviation of wind speed by average wind speed) is lower during high shear conditions and higher turbulence intensity is observed during low shear conditions. Similar model bias trends are observed when classifying the errors as a function of wind shear and turbulence intensity (not shown).




**Figure 8.** Distributions of stability parameter *z/L* during the (a) Humboldt and (c) Morro Bay deployments. Average (solid) and median (dashed) rotor-level wind speed bias according to *z/L* during the (b) Humboldt and (d) Morro Bay deployments, calculated over *z/L* intervals of length 0.1. Intervals with less than 10 samples are excluded.


## 3.3 Model Performance According to Wind Speed Class

At both buoy deployment locations, the models are found to slightly overestimate the slowest observed wind speed and strongly underestimate the fastest observed wind speeds (Fig. 9). At Humboldt, the models produce biases ranging from near zero (NARR) to 1.4 m s$^{-1}$ (MERRA-2) for observed wind speeds between 0 m s$^{-1}$ and 5 m s$^{-1}$. At Morro Bay, the models produce





biases ranging from near zero (RAP) to 1.1 m s$^{-1}$ (ERA5) for observed wind speeds between 0 m s$^{-1}$ and 5 m s$^{-1}$. Given typical turbine cut-in speeds around 3 m s$^{-1}$ to 5 m s$^{-1}$, this result indicates that the models may underrepresent the fraction of a wind farm lifecycle when the turbines are unable to produce power due to low wind speeds.

     Wind speeds between 5 m s$^{-1}$ and 10 m s$^{-1}$ represent the steepest portion of a typical turbine power curve and the models produce biases closest to 0 m s$^{-1}$ for this range (Fig. 9). At Humboldt, the model biases range from -1.5 m s$^{-1}$ (NARR) to 0.6
m s$^{-1}$ (ERA5) and at Morro Bay the model biases range from -1.3 m s$^{-1}$ (MERRA-2) to 0.1 m s$^{-1}$ (ERA5). Simulating power production using the National Renewable Energy Laboratory (NREL) 6 MW reference turbine (Musial et al., 2019), a wind speed bias of ±1 m s$^{-1}$ translates to discrepancies in gross capacity factor ranging from 8 to 23 percentage points when a turbine is operating at wind speeds between 5 m s$^{-1}$ and 10 m s$^{-1}$.

     At wind speeds near or at the top of typical turbine power curves (> 10 m s$^{-1}$), the models exhibit a trend of increasingly
negative bias with increasing observed wind speeds (Fig. 9). For all observed winds speeds exceeding 10 m s$^{-1}$ at Humboldt, the average model rotor-level wind speed biases range from -3.5 m s$^{-1}$ (NARR) to -0.3 m s$^{-1}$ (ERA5). For the fastest wind speed class at Humboldt, from 20 m s$^{-1}$ to 25 m s$^{-1}$, the biases range from -5.8 m s$^{-1}$ (NARR) to -1.7 m s$^{-1}$ (RAP). For all observed winds speeds exceeding 10 m s$^{-1}$ at Morro Bay, the average model rotor-level wind speed biases range from -4.0 m s$^{-1}$ (MERRA-2) to -1.3 m s$^{-1}$ (NARR). For the fastest wind speed class at Morro Bay, from 20 m s$^{-1}$ to 25 m s$^{-1}$, the biases
range from -7.0 m s$^{-1}$ (MERRA-2) to -3.1 m s$^{-1}$ (RAP). One of the reasons for large deviations within various models during high wind speeds could be due to unfavourable parameterization schemes used within the models for offshore conditions. For surface roughness calculations, MERRA-2 uses the Large and Pond (1981) parameterization scheme (Helfand and Schubert, 1995), which is known for significant deviations in offshore conditions (Edson et al., 2013). Since the surface roughness estimates are used within MOST for estimating the winds within the lowest few hundred meters, higher error in surface
roughness directly correlates to higher error in wind speed estimates. At very high wind speeds (greater than 25 m s$^{-1}$), surface roughness does not increase with wind speed and improper parameterizations can cause larger errors in models. Translating these findings to wind farm expectations, the large biases produce minimal discrepancies in simulated gross capacity factor when turbines are operating at wind speeds near the centre of the flat top portion of the power curve. The biases become significant for wind speeds near the edges of the flat top of the power curve, however, and can lead to misrepresentation of the
fraction of a wind farm lifecycle spent at peak production or beyond turbine cut-out.

     The MERRA-2, CFSv2, NARR, ERA5, and RAP biases between the slowest (0 m s$^{-1}$ – 5 m s$^{-1}$) and fastest (20 m s$^{-1}$ – 25 m s$^{-1}$) wind speed classes at Humboldt differ by magnitudes of 5.7 m s$^{-1}$, 4.9 m s$^{-1}$, 5.8 m s$^{-1}$, 3.9 m s$^{-1}$, and 2.0 m s$^{-1}$, respectively. The MERRA-2, CFSv2, NARR, ERA5, and RAP biases between the slowest and fastest wind speed classes at Morro Bay differ by magnitudes of 7.7 m s$^{-1}$, 5.5 m s$^{-1}$, 4.3 m s$^{-1}$, 6.3 m s$^{-1}$, and 3.1 m s$^{-1}$, respectively. Overall, larger biases are observed
at Morro Bay compared to Humboldt. Similar trends in bias, albeit smaller in magnitude, are observed while populating errors as a function of significant wave height.



**Figure 9.** Model wind speed bias according to observed wind speed during the (a) Humboldt and (b) Morro Bay deployments.

**3.4 Seasonal and Diurnal Trends in Model Error**

The accuracy of model representation of the observed California coast rotor-level winds varies according to season and time of day. The Morro Bay buoy provides observations over an entire seasonal cycle, identifying that the fastest rotor-level wind speeds occur between the months of November and June (Fig. 10f). Comparing the observations with the models reveals that MERRA-2, CFSv2, ERA5, and RAP strongly underestimate the observed lidar wind speeds between October and June by 2.0 m s$^{-1}$, 1.0 m s$^{-1}$, 0.8 m s$^{-1}$, and 1.0 m s$^{-1}$, respectively (Fig. 10g-j). During the warm period between July and September, which





is characterized by slow wind speeds (Fig. 10f), MERRA-2, CFSv2, and ERA5 produce less negative or even positive biases (-0.5 m s$^{-1}$, -0.1 m s$^{-1}$, and 0.8 m s$^{-1}$ on average, respectively), a transition that is particularly pronounced for ERA5. RAP tends

to produce consistently negative biases throughout the diurnal cycle between November and May at Morro Bay (-1.2 m s$^{-1}$) and biases closer to zero between June and October (-0.3 m s$^{-1}$).

The most pronounced diurnal patterns in model bias occur during the summer months at Morro Bay (Fig. 10g-j). At the buoy locations, 8 Coordinated Universal Time (UTC) corresponds to local midnight in Pacific Standard Time (PST) and 20 UTC corresponds to local noon in PST. Positive model rotor-level wind speed biases occur in the summer afternoons and

persist through the evening. The slowest wind speeds, shown to be correlated with positive wind speed bias (Fig.9), are present throughout the summer mornings and afternoons at Morro Bay, persisting into the evenings (Fig. 10f). This period also coincides with little to no wind shear and unstable atmospheric conditions.





**Figure 10.** (a), (f) Seasonal and diurnal 80 m wind speed at Humboldt and Morro Bay. Seasonal and diurnal wind speed bias when comparing (b), (g) MERRA-2 and observations at 50 m, (c), (h), the CFSv2 lowest 0-30 mb layer ASL and observations at 80 m, (d), (i) ERA5 and observations at 100 m, and (e), (j) RAP and observations at 80 m at Humboldt and Morro Bay.



At Humboldt, the models similarly show strong diurnal patterns in rotor-level wind speed bias during the warmer months for MERRA-2, CFSv2, and ERA5 (Fig. 10b-d), though a full seasonal analysis is limited due to data availability (Fig. 2a).

From June through October, MERRA-2, CFSv2, and ERA5 strongly overestimate the observed rotor-level wind speeds by 1.2 m s$^{-1}$, 0.9 m s$^{-1}$, and 1.2 m s$^{-1}$, respectively, from local noon to midnight PST. During the same months but for the hours after midnight through the morning, MERRA-2 and CFSv2 underestimate the observed wind speeds at Humboldt by 0.4 m s$^{-1}$ while ERA5 overestimates the observed wind speeds by 0.3 m s$^{-1}$. RAP shows a much subtler diurnal trend in rotor-level wind speed bias at Humboldt (Fig. 10e).

In order to visualize the trends in model error according to time of day at Humboldt and Morro Bay, the seasonal and diurnal patterns in rotor-level wind speed are presented in Fig. 11. A consistent diurnal pattern is present at Morro Bay throughout the year, with the fastest wind speeds occurring in the evening and at night and the slowest wind speeds occurring in the morning. All models capture the diurnal trend throughout the year at Morro Bay but consistently underestimate the observations. Inability of the models to predict the marine boundary layer depth can cause significant overestimation or underestimation of

winds within the region. Shallow marine atmospheric boundary layer depths are typically observed during summer months and could be one of the reasons for larger deviations between models and observations.

At Humboldt, the diurnal pattern in rotor level wind speed changes notably throughout the year. The diurnal patterns in modelled and observed wind speed at Humboldt align well for the month of November (Fig. 11m), with all sources showing the wind speed minimum occurring around local noon PST (20 UTC) and static winds throughout the evening and night.

Contrastingly, there is poor correlation between the modelled and observed diurnal wind speeds during the month of July at Humboldt (Fig. 11i). The lidar data show a pattern of static winds in the afternoon and evening that begin to increase just before local midnight PST (8 UTC) to a maximum around 3 PST (12 UTC), followed by a steady decrease throughout the morning hours (Fig. 10a, Fig. 11i). MERRA-2 shows the opposite diurnal pattern in July, with faster winds occurring during the afternoon and evening and slower winds occurring at night. RAP displays little variation in simulated wind speed according to the diurnal cycle, with a gentle minimum occurring in the afternoon. The CFSv2 wind show stronger variation according to

to the diurnal cycle, with a gentle minimum occurring in the afternoon. The CFSv2 wind show stronger variation according to the four daily runtimes of 0, 6, 12, and 18 UTC than according to the overall diurnal cycle. Like RAP, ERA5 displays little variation in wind speed according to time of day, with the exceptions of two sharp drops in the wind speed that occur 12 hours apart at 10 UTC (2 PST) and 22 UTC (14 PST).







**Figure 11.** Observed and modelled diurnal trends in wind speed at Humboldt and Morro Bay according to month.




## 3.5 Model Performance During West Coast Weather Phenomena

The period of October 2020 – September 2021 was not marked by many significant atmospheric or geologic events along the coast of California. The Humboldt and Morro Bay buoys were not impacted by any Pacific tropical storms or significant

earthquakes. The Morro Bay buoy was impacted by several weather events, wind reversals associated with the Santa Ana winds and an atmospheric river, and the following discussion provides insight on model wind speed performance during these events.

While the U.S. west coast is dominated by northerly and northwesterly winds (Fig. 1), abrupt reversals to southerly and southeasterly flow are known to occur. Bond et al. (1996) classify such wind reversals as coastally trapped, associated with

ageostrophic flow that is confined to the coastal zone, and synoptic, which are the result of landfalling frontal systems. The observations from the lidar buoys show drastically reduced rotor-level wind speeds at the central California Morro Bay site and slightly reduced rotor-level wind speeds at the northern California Humboldt site during southerly winds relative to the average speeds at each location (Fig. 12a, b), supporting the trend of increasing wind speeds with increasing latitude during reversals noted by Bond et al. (1996). Wind reversal events comprise 11% of the Morro Bay deployment and 23% of the

Humboldt deployment.

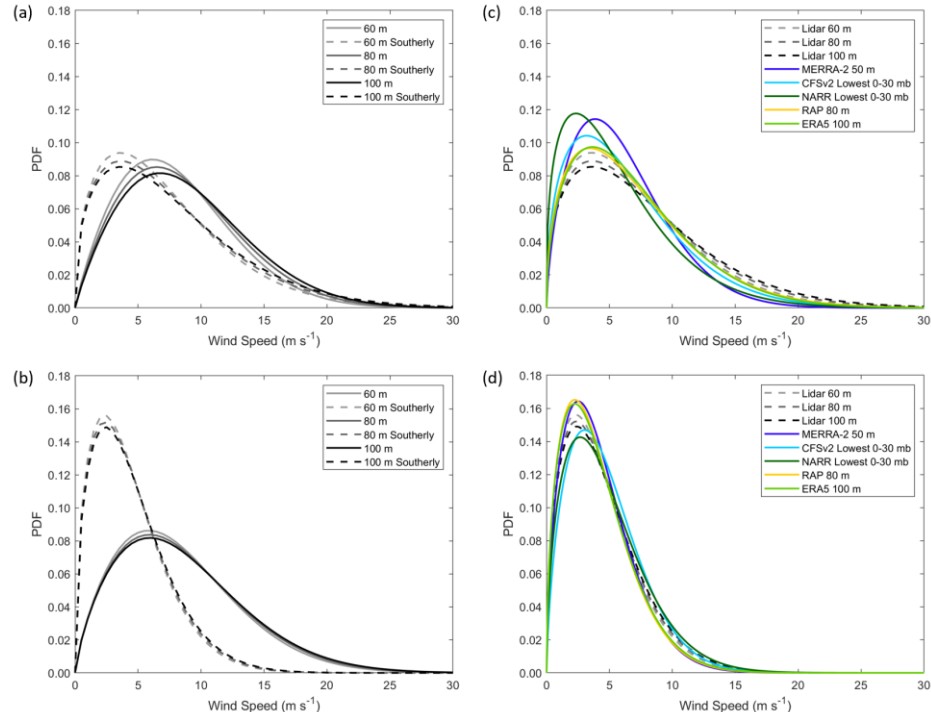

**Figure 12.** (a) Weibull fits to observed wind speeds for all wind directions (solid) and southerly flow (120°-240°) (dashed) at Humboldt. (b) Weibull fits to observed wind speeds for all wind directions (solid) and southeasterly flow (90°-210°) (dashed) at (b) Morro Bay. Weibull
fits to observed (dashed) and modelled (solid) wind speeds during southerly and southeasterly observed flow at (c) Humboldt and (d) Morro Bay, respectively.





The models underestimate the observed rotor-level wind speeds at the northern California Humboldt site during southerly wind events (Fig. 10a-e). MERRA-2, ERA5, and CFSv2 produce biases near -1 m s$^{-1}$ during wind reversals, which are larger than their deployment-wide biases of 0.0 m s$^{-1}$, 0.4 m s$^{-1}$, and -0.1 m s$^{-1}$, respectively. The RAP and NARR biases during wind reversals, -0.6 m s$^{-1}$ and -1.7 m s$^{-1}$, respectively, are more positive relative to their deployment-wide biases of -0.9 m s$^{-1}$ and -1.9 m s$^{-1}$, respectively.

At the central California Morro Bay site, all five models exhibit a reduction in rotor-level wind speed bias magnitude during wind reversals relative to their respective deployment-wide biases (Fig. 10f-j). MERRA-2, ERA5, and RAP underestimate the southerly observed rotor-level wind speeds during reversals with biases of -0.1 m s$^{-1}$, -0.4 m s$^{-1}$, and -0.3 m s$^{-1}$, respectively, reduced from their deployment-wide biases of -1.7 m s$^{-1}$, -0.4 m s$^{-1}$, and -0.8 m s$^{-1}$, respectively. CFSv2 and NARR overestimate the southerly observed rotor-level wind speeds at Morro Bay by 0.3 m s$^{-1}$ and 0.3 m s$^{-1}$, respectively, a sign transition from their deployment-wide biases of -0.8 m s$^{-1}$ and -0.3 m s$^{-1}$, respectively.

One meteorological phenomenon that can result in atypical wind directions along the central and southern California coasts are the Santa Ana winds. These warm, dry downslope winds are most frequently present in December and January (Guzman-Morales et al., 2016) and occur when a Great Basin high is present simultaneously with a surface low pressure system offshore (Raphael, 2003). This develops a counter-clockwise circulation zone offshore, creating a flow reversal at the buoy location. Four occasions of wind reversals during Santa Ana pressure setups are noted in the Morro Bay buoy record (Fig. 13).

The first event, 3 December 2020 15 UTC – 4 December 2020 0 UTC, is characterized by very low wind speeds (< 5 m s$^{-1}$). During this period, RAP is the best performing model at capturing the observed southerly winds at Morro Bay (Fig. 13b). ERA5 and CFSv2 remain northerly throughout the duration of the event, while MERRA-2 steadily transitions through the entire directional spectrum. The rotor-level wind speed biases during this event are small, ranging from -0.5 m s$^{-1}$ (RAP) to 0.2 m s$^{-1}$ (MERRA-2) (Fig. 13c, Table 3). NARR is excluded from the Santa Ana case study analysis due to the reduced sample size resulting from its coarser temporal resolution.

Low wind speeds similarly characterize the second event, 8 December 2020 11-23 UTC. RAP is the only model that accurately captures the southerly flow event (Fig. 13b). MERRA-2, CFSv2, and ERA5 remain northerly to northeasterly throughout the duration of the event. The rotor-level wind speed biases range from -0.8 m s$^{-1}$ (RAP) to 0.3 m s$^{-1}$ (CFSv2) (Table 3).

**Figure 13.** (a), (d) Observed wind direction, (b), (e) observed and modelled rotor-level wind direction, and (c), (f) observed and modelled rotor-level wind speed at Morro Bay.






Relatively higher wind speeds that correspond to the steep portion of a typical turbine power curve (4-9 m s$^{-1}$) are observed during the third event, 14 January 2021 13-23 UTC. Similar to prior events, RAP is successful at capturing the flow reversal (Fig. 13e), while the remaining models consistently show northerly flow. The rotor-level wind speed biases are more pronounced, ranging from -1.2 m s$^{-1}$ (MERRA-2) to -0.4 m s$^{-1}$ (RAP) (Table 3).


The fourth event, 17 January 2021 18-23 UTC, is characterized by low observed rotor-level speeds that follow a steep decline following a burst of wind exceeding 10 m s$^{-1}$ (Fig. 13f) associated with damaging winds along the central California coast (National Weather Service, 2022). None of the models capture the short-lived wind reversal and instead remain consistently northerly (Fig. 13e). The models substantially overestimate the observed rotor-level wind speeds, with biases ranging from 1.1 m s$^{-1}$ (RAP) to 3.5 m s$^{-1}$ (ERA5) (Table 3).


A powerful atmospheric river event impacted the western U.S. from 26-29 January 2021 (Weather Prediction Center, 2022). Heavy mountain snow and flooding were recorded across California. At the Morro Bay buoy, the recorded low pressure reached a minimum around the January 27-28 transition, an event that coincided with rapid shifts in the temperature, wind speed, and wind direction (Fig. 14).


From 13-21 UTC on January 27, the rotor-level winds transition from southerly to easterly flow, accompanied by wind speeds ranging from 9-13 m s$^{-1}$. At 22 UTC, the winds drastically shift to westerly and reduce to ~5 m s$^{-1}$. At 23 UTC, the wind direction begins to transition back to southerly flow and by midnight UTC on January 28, the rotor-level wind speeds spike to 22 m s$^{-1}$. The enhanced wind speeds sustain for a period of three hours before rapidly decreasing to ~6 m s$^{-1}$.

MERRA-2 and CFSv2 fail to capture the rapid changes in wind direction, remaining southerly throughout the duration of the event. ERA5 captures the initial shift to easterly winds, albeit several hours early, but subsequently remains southerly throughout the remainder of the event, missing the drastic shift to westerly winds. Only RAP captures the entire transition from southerly to easterly to westerly and back to southerly winds and within ±1 hour of accuracy. In terms of wind speed, only RAP captures the rapidly changing wind pattern throughout the event, though with a low bias of -2.2 m s$^{-1}$ (Table 3).


MERRA-2, CFSv2, and ERA5 fail to capture the rapid wind speed transitions, instead producing gently sloped peaks in the modelled wind speed that occur before (MERRA-2) or after (CFSv2, ERA5) the observed wind speed peak. This mischaracterization leads to substantial overestimation during the atmospheric river event, with rotor-level wind speed biases of 3.5 m s$^{-1}$, 3.2 m s$^{-1}$, and 1.7 m s$^{-1}$ for MERRA-2, CFSv2, and ERA5, respectively (Table 3).




**Figure 14.** (a) Observed pressure, (b) observed air and sea temperature, (c) observed wind speed, and (d) observed and modelled wind speed at the Morro Bay buoy during the January 2021 western U.S. atmospheric river event.





**Table 3.** Model bias during Pacific coast weather events during the lidar buoy deployments.

| Event | Site | Duration (UTC) | MERRA-2 Bias | CFSv2 Bias | RAP Bias | ERA5 Bias |
|---|---|---|---|---|---|---|
| Santa Ana Winds | Morro Bay | 3 Dec 2020 15 <br> 4 Dec 2020 00 | 0.2 m s$^{-1}$ | -0.1 m s$^{-1}$ | -0.5 m s$^{-1}$ | -0.1 m s$^{-1}$ |
| Santa Ana Winds | Morro Bay | 8 Dec 2020 11 <br> 8 Dec 2020 23 | -0.3 m s$^{-1}$ | 0.3 m s$^{-1}$ | -0.8 m s$^{-1}$ | -0.7 m s$^{-1}$ |
| Santa Ana Winds | Morro Bay | 14 Jan 2021 13 <br> 14 Jan 2021 23 | -1.2 m s$^{-1}$ | -1.0 m s$^{-1}$ | -0.4 m s$^{-1}$ | -0.6 m s$^{-1}$ |
| Santa Ana Winds | Morro Bay | 17 Jan 2021 18 <br> 17 Jan 2021 23 | 1.5 m s$^{-1}$ | 2.9 m s$^{-1}$ | 1.1 m s$^{-1}$ | 3.5 m s$^{-1}$ |
| Atmospheric River | Morro Bay | 27 Jan 2021 13 <br> 28 Jan 2021 07 | 3.5 m s$^{-1}$ | 3.2 m s$^{-1}$ | -2.2 m s$^{-1}$ | 1.7 m s$^{-1}$ |

### 3.6 Model Performance in Capturing Ramp Event Frequency

Ramp events, large changes in power production over relatively short temporal scales that are critical for power system management (on the order of a few hours), are one of the main challenges for the operation of power systems with significant contributions from wind power (Drew et al., 2018; Valldecabres et al., 2020). Rapid and unexpected increases in wind speed can lead to up ramps in power, which can lead to grid overload. Sudden reduction in the wind resource can result in the need to quickly rebalance the power supply with alternative sources. Ramp events occur for a variety of reasons, including the passage of frontal systems and turbine cut-out due to extreme wind speeds (Drew et al., 2018). Accurate representation of ramp events in numerical weather prediction models is essential for reducing the uncertainty associated with wind in power systems.

Wind ramps are defined by the change in power $\Delta P$ that occurs over a duration of time $\Delta T$, and the thresholds for these parameters vary widely across the industry. By combining observed and reanalysis wind speeds with the NREL 6 MW reference power curve (Musial et al., 2019), we obtain timeseries of power $P$. Prior to conversion to power, the wind speeds $v$ are normalised for air density $\rho$ using the near surface temperature and pressure observations according to IEC (2005), where the power curve standard density $\rho_0 = 1.225$ kg m$^{-3}$:

$$v_n = v \left( \frac{\rho}{\rho_0} \right)^{1/3} \tag{6}$$





Based on the wind ramp literature review of Gallego-Castillo et al. (2015), ramps in this study are identified using the following commonly utilised scheme which allows for classification of the sign of the ramps (positive $\Delta P$ indicates an up ramp, while negative $\Delta P$ indicates a down ramp):

$$\Delta P = P_{T+\Delta T} - P_T \tag{7}$$

The changes in power $\Delta P$ are represented as a percent change relative to the turbine rated capacity (6 MW, in this example).

Given the temporal output frequency of the models, we analyse ramps of durations $\Delta T$ ranging from one to three hours. It is important to note that ramps of shorter duration may also contribute to ramps of longer duration, i.e., a ramp with a 40% change in power over one hour might be part of a ramp with a 60% change in power over two hours.

During the Humboldt deployment, 2192 wind ramps with $\Delta P$ ranging from 30% to 100% of rated capacity are identified based on wind speed observations at 100 m (Fig. 15a). Using the observational wind speeds at 50 m and 80 m, the number of

ramp occurrences reduces to 1793 and 1910, respectively. During the Morro Bay deployment, 2558 wind ramps are identified based on wind speed observations at 100 m (Fig. 15b). The observational wind speeds at 50 m and 80 m produce 2299 and 2442 ramp occurrences, respectively.

As in Bianco et al. (2016), ramps characterized by small changes in power over longer durations are predominant (Fig. 15). At both Humboldt and Morro Bay, roughly equal proportions of up and down ramps occur, in contrast to the East Coast buoy

deployments which identified more frequent up ramps (53%) versus down ramps (Sheridan et al., 2021).

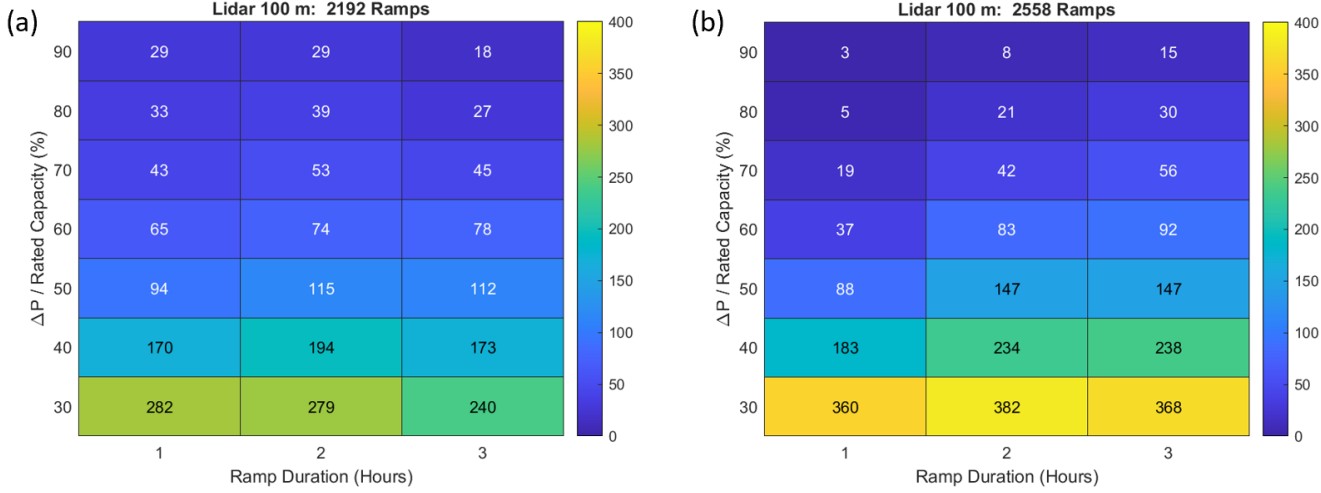

**Figure 15.** Combined up and down ramp frequency during the (a) Humboldt and (b) Morro Bay deployments using lidar observations at 100 m ASL and the NREL 6 MW reference power curve (Musial et al., 2019).

In most cases, the models underestimate the number of ramp events identified using the West Coast lidar buoy observations

(Fig. 16). The coarsest reanalysis, MERRA-2, produces only 17% and 16% of the observed 50 m ramps at Humboldt and Morro Bay, respectively. Next, ERA5 produces 29% and 41% of the observed 100 m ramps at Humboldt and Morro Bay, respectively. The model with the highest spatial resolution, RAP, captures 50% and 61% of the observed 80 m ramps at



Humboldt and Morro Bay, respectively. In contrast, CFSv2 overestimates the number of observed 80 m ramps at Humboldt by 135%. At Morro Bay, CFSv2 captures 71% of the observed 80 m ramps.

At Humboldt, none of the models produces the 50% ratio of observed up ramps to down ramps. MERRA-2 strongly favours up ramps (62%), while ERA5 and RAP slightly favour up ramps (55% and 53%, respectively). CFSv2 produces more down ramps (53%) than up ramps. At Morro Bay, CFSv2 produces the 50% ratio of observed up ramps to down ramps. As at Humboldt, MERRA-2 at Morro Bay strongly favours up ramps (70%), while ERA5 and RAP slightly favour up ramps (55% and 52%, respectively).

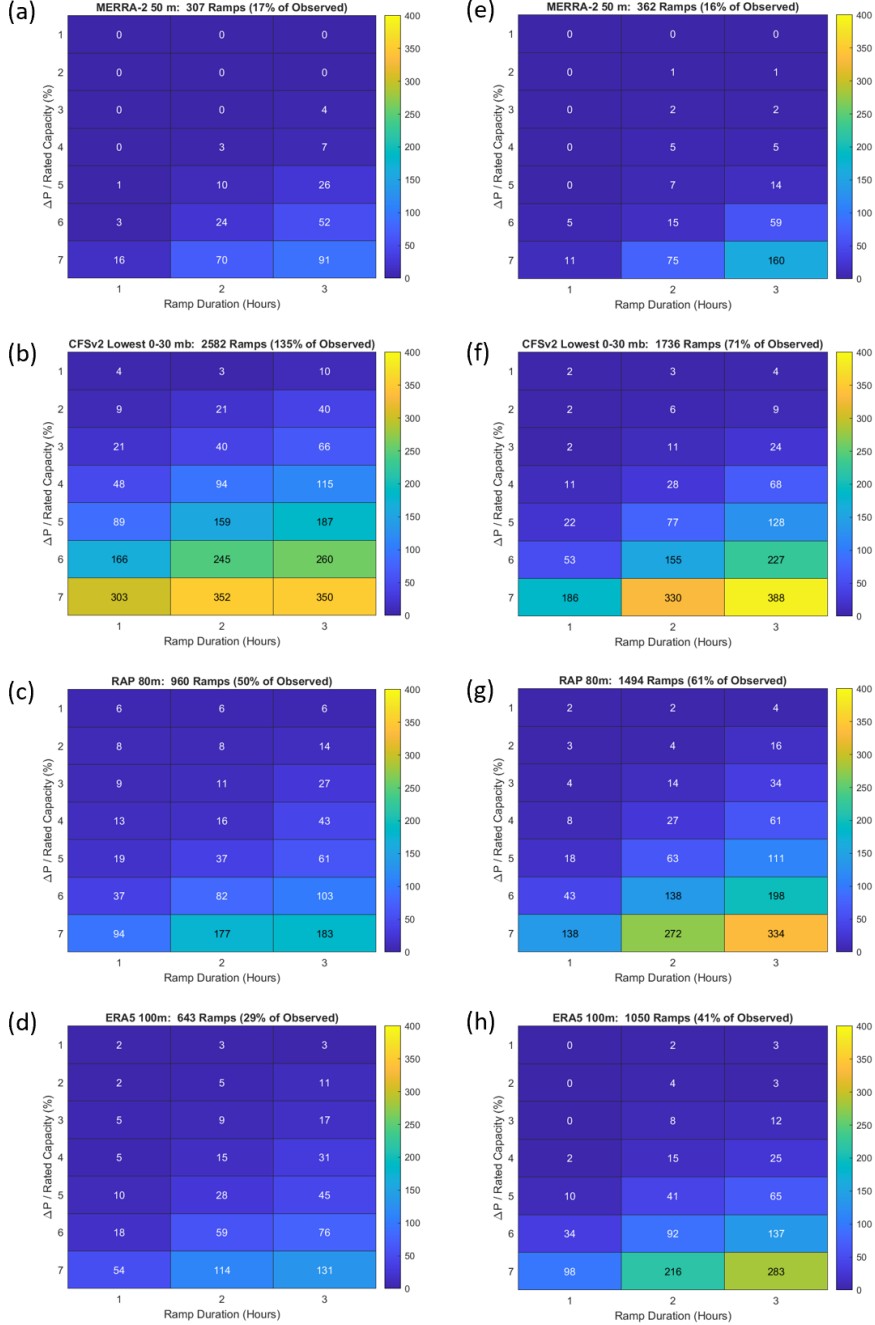


**Figure 16.** Combined up and down ramp frequency during the Humboldt deployment using (a) 50 m MERRA-2 wind speeds, (b) the lowest 0-30 mb CFSv2 wind speeds, (c) 80 m RAP wind speeds, (d) 100 m ERA5 wind speeds and the NREL 6 MW reference power curve (Musial et al., 2019). Combined up and down ramp frequency during the Morro Bay deployment using (e) 50 m MERRA-2 wind speeds, (f) the lowest 0-30 mb CFSv2 wind speeds, (g) 80 m RAP wind speeds, (h) 100 m ERA5 wind speeds and the NREL 6 MW reference power curve (Musial et al., 2019). Percentages of observed ramps are compared using the observations at the same height as the model output (i.e., the number of RAP ramp occurrences at 80 m is compared with the number of ramp occurrences produced using the 80 m lidar wind speed observations).



## 4 Conclusions

As offshore wind plays an increasing role in the U.S. and global energy portfolios, the need to validate the models that support
long-term wind resource characterization in areas of offshore wind development interest similarly increases. The DOE lidar
buoys provide essential observations for validation at the spatial scales most relevant for offshore wind: over the water and at
rotor layer height. The recent deployments of the lidar buoys off the coast of California in two areas of offshore wind
development interest, Humboldt and Morro Bay, reveal a trend of model underestimation of the observed hub height wind
speeds. At the northern California Humboldt location, MERRA-2 and CFSv2 yield the smallest rotor-level wind speed biases,
near zero, while NARR produces the largest magnitude bias of -1.9 m s$^{-1}$. Contrastingly, at the central California Morro Bay
location, MERRA-2 produces the largest magnitude rotor-level wind speed bias of -1.6 m s$^{-1}$, while NARR and ERA5 yield
the smallest biases of -0.3 m s$^{-1}$ and -0.4 m s$^{-1}$, respectively.

For the direct model level to lidar level comparisons, RAP, the model with the highest spatial resolution, provides the lowest
CRMSEs (2.3 m s$^{-1}$ at Humboldt, 1.7 m s$^{-1}$ at Morro Bay) and the highest correlations (0.88, 0.94). MERRA-2, the coarsest
model, produces the highest CRMSEs (2.7 m s$^{-1}$, 2.6 m s$^{-1}$) and lowest correlations (0.79, 0.86). The lowest 0-30 mb layer
ASL from CFSv2, NARR, and RAP provides an appropriate representation of 80 m wind speeds based on correlations of 0.82
or greater along with biases and CRMSEs of similar magnitude to those produced based on the direct level comparisons.

An investigation into the conditions leading to large simulation error reveals model mishandling of the summer diurnal
pattern in the wind speed at the northern Humboldt location, though the diurnal pattern in cooler months is well captured.
Trends in reanalysis wind speed bias according to atmospheric stability are location-dependent, with model underestimation
of the observed wind speeds during near neutral conditions at both sites, but overestimation of the observed winds during
stable conditions at Humboldt and unstable conditions at Morro Bay. Model bias varies strongly according to observed wind
speed class and weakly according to significant wave height. At Humboldt, MERRA-2, ERA5, and CFSv2 produced larger
magnitude biases during wind reversal events relative to the deployment-wide biases, while at Morro Bay all models
experience a reduction in bias magnitude during wind reversals. For wind reversals at Morro Bay resulting from the Santa Ana
winds, RAP is the best performing model at capturing directional shifts. All models yield simulated ramp event frequencies
significantly different than the frequencies suggested by the lidar observations, with CFSv2 the most successful model in this
metric.

The upcoming effort for continuing this research begins with the re-examination of model performance at Humboldt once
an entire seasonal cycle is available at that location. Next, model performance can be analysed according to additional
atmospheric and oceanic phenomena of interest to the wind energy community, such as low-level jets (observed at Morro Bay
buoy location). Understanding the physical processes leading to the changing biases in the diurnal cycle seen throughout the
year will be important for guiding use of both reanalyses and understanding forecasts for these offshore wind locations. Finally,
the observations from the California deployments of the DOE lidar buoys will be used to validate the performance of coupled
wind/wave simulations of offshore winds (Gaudet et al., 2022).



**Code and Data Availability**

The data utilised in this study are freely and publicly available. The lidar buoy observations are available from the U.S. Department of Energy at a2e.energy.gov. NASA provides MERRA-2 through the Goddard Earth Sciences Data and Information Services Center at https://gmao.gsfc.nasa.gov/reanalysis/MERRA-2/data_access/. CFSv2 and NARR are accessed via Research Data Archive provided by the National Center for Atmospheric Research at rda.ucar.edu. ERA5 is available through the Copernicus Climate Change Service Climate Data Store at cds.climate.copernicus.eu. RAP is available for download at NOAA's National Centers for Environmental Information at ncei.noaa.gov. Data from neighbouring buoys are provided by NOAA's National Data Buoy Center at ndbc.noaa.gov. Satellite data from the collection of Ribal and Young (2019) are available at http://thredds.aodn.org.au/thredds/catalog/IMOS/SRS/Surface-Waves/Wave-Wind-Altimetry-DM00/catalog.html.

For convenience, paired timeseries of wind data from the model and observations at the Humboldt location are provided at https://a2e.energy.gov/data/buoy/reanalysis.z05.c0 (DOI: 10.21947/1839076), along with processing scripts. Paired timeseries of wind data from the model and observations at the Morro Bay location are provided at https://a2e.energy.gov/data/buoy/reanalysis.z06.c0 (DOI: 10.21947/1839076), along with processing scripts.

**Author Contributions**

LS accessed and processed the data and led the data analysis with significant contributions from RK, GGM, and BG. RK, AM, and WS led the project and provided essential direction for this research. WG and ZY provided additional valuable feedback and guidance for the project. MP and RN were instrumental in the acquisition and processing of the lidar buoy observations.

**Competing Interests**

The authors declare that they have no conflicts of interest.

**Acknowledgements**

This work was authored by Pacific Northwest National Laboratory, operated for the U.S. Department of Energy by Battelle under contract DE-AC05-76RL01830. Additionally, the authors would like to thank Shannon Davis and Mike Derby at the U.S. Department of Energy Wind Energy Technologies Office for funding this research and Doug Boren, Necy Sumait, and Frank Pendleton at the U.S. Department of the Interior Bureau of Ocean Energy Management for funding the Humboldt and Morro Bay deployments. The authors would also like to thank Julia Flaherty and Catie Himes for reviewing the preliminary draft of the article.



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
