# Peer review of "Offshore Reanalysis Wind Speed Assessment Across the Wind Turbine Rotor Layer off the United States Pacific Coast"

_Wind Energy Science, 2022_

## Author Comment (AC1)

The manuscript by Sheridan et al. compares wind data from several numerical products against floating lidar buoy observations from two sites at along the US Pacific Coast. This is relevant and interesting in particular because of the future relevance of the region for offshore wind energy utilization. This is also the main strength, why I like the manuscript not really the scientific novelty.

The manuscript is in general well written and readable. However, I have several major and minor aspects to be clarified before I can recommend publication in Wind Energy Science. This applies to the representation of the figures, interpretation of the results and description of underlying mechanisms.

The authors are very grateful for your time in reviewing our paper and for the helpful recommendations you provided. We have worked to address the aspects of the paper requiring improvement per your suggestions as follows.

**Major Points:**

1. **Readability of panel figures.** Later in the document there are several panel figures where the font size is very small and thus hard to read. As these are partly showing same quantities there are ways to increase figures sizes by e.g. sharing axes. This is for example the case for Figures 5, 10, 11, 12
Thank you for the suggestion to improve the visibility of the figures. Figure 5 has been updated with larger font size. Figure 10 was split into two figures (11 and 12) to improve clarity. Figure 11 was also split into two figures (13 and 14) to improve clarity. Figure 12 (now Figure 15) has been updated with larger font size.

2. **Discussion of results:** There were several points in the presentation of the results where a phenomenon is correctly described but the interpretation and discussion is much to short or completely missing.
Thank you for bringing this to our attention. We have supplemented the short/missing interpretations you pointed out as follows:

A few examples:

- Line 273 You mention that RAP is the best performing model at both sites. But there is no reason / interpretation given why.
We have added the following sentence to our statement about RAP on Line 293:

   "The high spatial resolution of RAP is a likely factor in the model success, providing ability to represent features and phenomena that the coarser models miss, as is explored in upcoming sections. It is also possible that the relatively rapid analysis cycle of RAP (hourly) versus some of the reanalyses may be a factor."

- Line 301: The two sites tend to different stabilities. Why is this the case? No reason is given. Or is it just the different availability (see major comment #3)
  Per your suggestion, we have considered the overlapping period of data recovery between the two locations and have found that the tendency toward different stabilities (Humboldt near neutral, Morro Bay unstable) holds. We have added the following reasoning to the text, beginning on Line 329:

  "A contributing factor for the difference in stability between the two locations is the air-sea temperature differential. At the northern Humboldt location, the average air temperature and sea surface temperature during the deployment are both 12.0°C. Further south at Morro Bay, the average air and sea surface temperatures increase as expected, however not equally. The average air and sea surface temperatures at Morro Bay during the overlapping Humboldt deployment period are 13.9°C and 14.6°C, respectively. The trend of cooler air relative to warmer ocean results in more frequent unstable conditions."

[Figure]

- Line 321: Where do I see strong gradients in boundary layer depths? In what figure/reference? Or is this just guessing?
  Thank you for pointing out that we are missing information here. We have added the following references to the sentence beginning on Line 354:

  Ao, C. O., Waliser, D. E., Chan, S. K., Li, J-L., Tian, B., Xie, F., Mannucci, A. J., Planetary boundary layer heights from GPS radio occultation refractivity and humidity profiles, Journal of Geophysical Research-Atmospheres, 117, D16117, https://doi.org/10.1029/2012JD017598, 2012.

  Ström, L. and Tjernström, M., Variability in the summertime coastal marine atmospheric boundary-layer off California, USA, Quarterly Journal of the Royal Meteorological Society, 130, 423-448, https://doi.org/10.1256/qj.03.12, 2004.

Juliano, T. W., Parish, T. R., Rahn, D. A., and Leon, D. C., An Atmospheric Hydraulic Jump in the Santa Barbara Channel, Journal of Applied Meteorology and Climatology, 56, 11, 2981-2998, https://doi.org/10.1175/JAMC-D-16-0396.1, 2017.

- Line 337: Models are found to slightly overestimate the slowest wind speed and strongly underestimate the fastest wind speed. What is the reason for this?
  In order to enhance the discussion, the following paragraph beginning on Line 419 has been added to the section examining model error according to observed wind speed:

  "There could be several compounding reasons for such inconsistencies. It is likely that the model underestimation of wind speed in the highest observed wind speed class is due to the coarse resolution of the models and the small spatial scale of these features. The reason for the model overestimation for the lowest observed wind speed class is less clear, but it could be related to unresolved wind direction variability in low-speed conditions (which would tend to reduce the observed vector-averaged wind speed), or to biases in the model parameterization of wave roughness in low speed conditions.

- Line 410: Shallow ABL are described to be observed during summer months. Why is this the case? This is especially important as the opposite is true onshore
  We have added the following discussion and reference to the manuscript on Line 469 to explain the occurrence of shallow marine boundary layer depths in summer months:

  "Larger atmospheric boundary layer depths tend to be associated with more positive surface heat fluxes to the atmosphere, which are in turn associated with large surface-air temperature differentials. Over land, this is primarily driven by the response of surface temperature to local solar heating, which is more intense in summer. Over the ocean, however, the local change in air temperature in the summer tends to be greater than the ocean temperature change, which may even be negative when upwelling is present. Therefore, in summer the atmospheric stability actually tends to be increased relative to winter, leading to less positive (or even negative) surface heat fluxes and shallower atmospheric boundary layer depths. The summertime position of the North Pacific subtropical high pressure causes the marine atmospheric boundary layer to slope downward near the taller mountainous California coast, trapping the boundary layer between the surface, the mountains, and the inversion (Dorman et al., 1999; Ström and Tjernström, 2004). The inversion base in this region can range from 100 m to 800 m during the summer, but typically is observed between 300 m and 400 m (Dorman, 1985, 1987; Juliano et al., 2017)."

- Line 465: model performance is described differently but no reasons are given why this is the case? Is it resolution? Assimilation of different data?
  We attribute the distinctions in model performance to the distinctions in model spatial resolution. Of our four wind reversal case studies, only RAP, the model with the highest spatial resolution, was found to capture the reversals in three of the four events. We have amended the text on Line 537 to present this attribution:

  "During this period, RAP is the best performing model at capturing the observed southerly winds at Morro Bay (Fig. 16b), likely due to its high spatial resolution."

- Line 484: None of the models can capture the wind reversal. Is that such a small scale phenomenon that they all don't resolve?
  In the four wind reversal case studies presented in the paper, RAP is the best performing model in that it captures three of the four reversal events, while the remaining models do not capture any of the events.

  The wind reversal event that is not captured by any of the models, including RAP, is unique compared to the other three events in that it follows a rapid increase and subsequent rapid decrease in the observed wind speed, which the models similarly do not resolve.

  The discussion beginning on Line 554 has been amended as follows: "The fourth event, 17 January 2021 18-23 UTC, is characterized by low observed rotor-level speeds that follow a steep decline following a burst of wind exceeding 10 m s$^{-1}$ (Fig. 16f) associated with damaging winds along the central California coast (National Weather Service, 2022). None of the models capture the short-lived wind reversal and instead remain consistently northerly (Fig. 16e). The models substantially overestimate the observed rotor-level wind speeds, with biases ranging from 1.1 m s$^{-1}$ (RAP) to 3.5 m s$^{-1}$ (ERA5) (Table 3). Notably, none of the models capture the observed burst of wind and the subsequent steep decline in the observed wind speed prior to the wind reversal event, indicating model challenges in resolving atmospheric phenomena that change over periods of short duration."

Please go through the whole manuscript and especially the results section and improve the document with respect to this.

3. **Comparison of Humboldt and Morro Bay:** I think it is fair to show results for two different sites but a comparison of model performance for the two sites is just not possible at all. There is a distinct annual cycle in the wind condition along the coastline and the measurements do not overlap in time. Please remove any comparisons of the two sites from the manuscript that cover data from periods where the Humboldt measurement data weren't available.
  The authors greatly appreciate the reviewer pointing out that comparing the two sites while they have different data coverage periods is not recommended. To address this suggestion and to add value, we have provided the following metrics throughout the paper:
  1. Metrics at Humboldt for the Humboldt data coverage periods of 1 October 2020 – 27

December 2020 and 25 May 2021 – 30 September 2021

2. Metrics at Morro Bay during the overlapping Humboldt data coverage periods of 1 October 2020 – 27 December 2020 and 25 May 2021 – 30 September 2021, in order to compare with Humboldt

3. Metrics at Morro Bay for the full deployment period of 1 October 2020 – 30 September 2021, in order to examine the full seasonal cycle at this location

Line 97 now states: "In this article, the period of study for both buoys is 1 October 2020 to 27 December 2020 and 25 May 2021 to 30 September 2021. Additionally, the full seasonal cycle of 1 October 2020 to 30 September 2021 is analysed for Morro Bay."

Metrics for each of these three considerations are presented throughout the entire manuscript, as the following example from the conclusions section on Line 595 shows:

"At the northern California Humboldt location, MERRA-2 and CFSv2 yield the smallest rotor-level wind speed biases, near zero, while NARR produces the largest magnitude bias of -1.9 m s$^{-1}$. Contrastingly, at the central California Morro Bay location during the overlapping Humboldt period, MERRA-2 produces the largest magnitude rotor-level wind speed bias of -1.3 m s$^{-1}$ (-1.6 m s$^{-1}$ during the full Morro Bay deployment), while NARR and ERA5 yield the smallest biases of 0 m s$^{-1}$ (-0.3 m s$^{-1}$) and -0.2 m s$^{-1}$ (-0.4 m s$^{-1}$), respectively."

Figures have been updated throughout the text to provide both the overlapping period between the two buoys and the full Morro Bay seasonal cycle, as the following example shows:

[Figure]

4. **Ramp Events:** Ramp events are typically considered on time scales below 1h. I don't understand the motivation for investigating hourly data here. The thread to the grid is much lower on these time scales as with a deep penetration of wind energy there be balancing between plants that are apart. What are the underlying mechanisms that the models resolve? Ramp events are e.g. in the North Sea often found for mesoscale phenomena that the reanalysis datasets aren't resolving. In the current version, I don't see the benefit of this section at all.

   Thank you for the suggestion. We agree that the model temporal resolution does not allow for the most useful ramp event analysis and have therefore removed the section.

**Minor Points:**

- Figure 1: The color of the wind farm areas are similar to the terrain color, please chose a different one

  We appreciate the suggestion for this figure and have recreated it with the wind energy areas highlighted in a more distinct color.

- Line 100: I think this sentence fits better at the beginning of section 2

  Thank you for the suggestion. We have moved the sentence to the beginning of Section 2.

- ERA5: 0.5 degrees. ERA5 is provided in 0.25 degree resolution. What is the reason for the averaging here?

  0.5 degrees is a typo. The ERA5 resolution incorporated in the study was 0.25 degree. This typo has been fixed in Table 2. Thank you!

- Line 170: Is the daily cycle covered with these satellites? I guess the overpass is always at the same time of the day?

  The satellite reporting hours in our areas of interest span the diurnal cycle throughout the analysis period, but no individual day provides full representation of the diurnal cycle. For example, at the Humboldt location, the satellites report on 1 October 2020 at 18:43 UTC and then the next reports are 6 October 2020 at 5:49 UTC, 7 October 2020 at 2:36 UTC, 11 October 2020 at 16:41 UTC, and so on.

  We have modified the discussion beginning on Line 181 to the following to provide clarity per your helpful question: "627 and 203 satellite data points are collected between October 2020 and September 2021 within a 30-km radius of the Humboldt and Morro Bay buoys, respectively, at times spanning the diurnal cycle, though no individual days provide complete representation of the diurnal cycle."

- Line 226: underestimates the average at 50m by 1.6m/s. I guess you mean -1.6 m/s?

  Thank you for pointing out that this sentence needs clarification. We have reworded Line 233 in the updated text accordingly: "MERRA-2, the coarsest of the models, strongly underestimates the average observed wind speed at 50 m ASL with a bias of -1.6 m s$^{-1}$."

- Line 313: The impact of stability is also commonly observed in MOST, which typically shows larger errors during stable conditions -> I don't understand this sentence at all, why observed?

  We have improved the sentence on Line 345 for clarity in the following way and have

included a reference: "The accuracy of using MOST for wind profile extrapolation varies according to atmospheric stability, with larger errors typically occurring during stable atmospheric conditions (Optis et al., 2015)."

Optis, M., Monahan, A., and Bosveld, F. C., Limitations and breakdown of Monin-Obukhov similarity theory for wind profile extrapolation under stable stratification, Wind Energy, 19, 6, 1053-1072, https://doi.org/10.1002/we.1883, 2015.

- Line 327: Similar model bias trends are observed when classifying as function of wind shear and turbulence intensity. As these ones, especially the shear one are much easier to measure with lidars than stability or turbulence, I think it is valuable to also show this, even when results are quite similar.
  Thank you for this helpful suggestion. We have added a discussion on the model performance trends according to wind shear in Section 3.3.

- Line 360: ... at very high wind speeds surface roughness does not increase with wind speed.... Do you have a reference for this? Isn't the surface roughness offshore increasing with increasing wind speed, e.g. due to wave breaking?
  Here is the reference, and we have included it in the manuscript on Line 412:

  Donelan, M. A., Haus, B. K., Reul, N., Plant, W. J., Stiassnie, M., Graber, H. C., Brown, O. B., and Saltzman, E. S., On the limiting aerodynamic roughness of the ocean in very strong winds. Geophysical Research Letters, 13, 18, https://doi.org/10.1029/2004GL019460, 2004.

- Line 365: ... or beyond turbine cut-out... Many offshore wind turbines nowadays derate the power slowly after a certain threshold which is more grid friendly. This should be mentioned somewhere as this is relevant for e.g. the ramping discussions.
  Great point! We've updated the discussion to the following on Line 414: "The biases become significant for wind speeds near the edges of the flat top of the power curve, however, and can lead to misrepresentation of the fraction of a wind farm lifecycle spent at peak production. The biases can also lead to misrepresentation of the portion of a wind farm life cycle spent beyond turbine cut-out or in the state when turbine power slowly derates during very high wind speeds, depending on the turbine technology."

- Line 439: Wind reversal events comprise 11% of Morro Bay and 23 % of Humboldt deployment. This is such an example where a comparison is just not possible and shouldn't be made due to the different measurement periods.
  We agree and have updated the sentence on Line 509 as follows to compare during the overlapping periods: "Wind reversal events comprise 11% of the entire Morro Bay deployment and 11% and 22% of the Morro Bay and Humboldt deployments, respectively, during the overlapping data recovery periods of 1 October 2020 – 27 December 2020 and 25 May 2021 – 30 September 2021."

---

## Author Comment (AC2)

The authors' work aims at providing more reliable data for the analysis of offshore wind resources – a generally very welcoming undertaking. Likewise is the attempt of making more – and better – use of reanalysis data for that purpose also highly appreciated.

However, as always when using model output, validating the results is a necessary but many times difficult process. The lack of sufficient in-situ measurements of offshore wind speeds makes this an especially challenging task.

Now, how do the authors tackle this problem? The manuscript is kind of a mixed bag. I did not identify any technical errors – that's good news. But the nature of the manuscript is very descriptive, consisting mainly of 'naked' results of comparison/deviations of the different data sets. This is in part due to the fact that before making thorough meteorological analyses the data have to be spread out and subjected to a first statistical analysis. Here, I acknowledge the authors' achievement of providing a lot of data for comparisons of a few offshore wind measurements with results of most of the available reanalysis models. This is indeed a bulk of work and – to my knowledge – has not been done before that extensively. So, the first step has been made. The other side of the coin is a substantial lack of interpretation/analysis of the results with respect to the physical processes determining these results. This however is necessary in order to make use of the authors' results in a more generalised manner.

So, in the end my recommendation is to accept the paper as it is having in mind that the work presents a very good basis for further investigations but is still incomplete. The authors are encouraged to proceed to the next stage of interpreting the results and provide meteorological explanations to the partially very large discrepancies between measurements and modelling results. This is what the scientific community requires.

The authors are extremely appreciative of your time spent in reviewing our manuscript and providing feedback. We agree with your assessment of what the work discussed in this paper is intended to be: a quantification of model errors in two areas of relevant offshore wind development potential and identification of meteorological conditions leading to large model error. The sources of error in reanalysis models could vary due to different model grid resolution, vertical extrapolation errors due to Monin-Obukhov similarity theory, appropriate choice of climate models in various reanalysis, planetary boundary layer schemes chosen within each model, and different data assimilation data types in each reanalysis. Therefore, it is rather a grand challenge to decipher some of these compounding effects on reanalysis model errors when compared to observations. We have added some additional text in the current version of the manuscript providing additional justification to the results, where possible.

The recommended further investigations to interpret the results on a deeper level and use them to improve models are a part of our current research efforts. These approaches are being taken from a modeling standpoint, exploring the improvement of accuracy provided by higher resolution models that better capture meteorological phenomena, coupled wind-wave models, and the selection of appropriate planetary boundary layer schemes.

We have performed similar stages of research using the lidar buoys on the U.S. Atlantic coast (quantification/identification of model errors followed by a deeper investigation to try and understand/improve such discrepancies). We invite the reviewer to check out the deeper investigation we provided for the Atlantic analysis to gain a preview of our ongoing research efforts in the Pacific:

Gaudet, B.J., García Medina, G., Krishnamurthy, R., Shaw, W. J., Sheridan, L. M., Yang, Z., Newsom, R. K., and Pekour, M., Evaluation of coupled wind / wave model simulations of offshore winds in the Mid-Atlantic Bight using lidar-equipped buoys, Monthly Weather Review, 150, 6, 1377-1395,  https://doi.org/10.1175/MWR-D-21-0166.1, 2022.